# MS$^3$M: Multi-Stage State Space Model for Motion Forecasting

## Abstract

Motion forecasting is a fundamental component of autonomous driving systems, as it predicts an agent's future trajectories based on its surrounding environment. Transformer architectures have dominated this domain due to their strong ability to model both temporal and spatial information. However, transformers often suffer from quadratic complexity with respect to input sequence length, limiting their ability to efficiently process scenarios involving numerous agents. Additionally, transformers typically rely on positional encodings to represent temporal or spatial relationships, a strategy that may not be as effective or intuitive as the inductive biases naturally embedded in convolutional architectures. To address these challenges, we leverage recent advancements in state space models (SSMs) and propose the Multi-Stage State Space Model (MS$^3$M). In MS$^3$M, the Temporal Mamba Model (TMM) is employed to capture fine-grained temporal information, while the Spatial Mamba Model efficiently handles spatial interactions. By injecting temporal and spatial inductive biases through Mamba's state-space model structure, the model's capacity is significantly improved. MS$^3$M also strikes an exceptional trade-off between accuracy and efficiency, which is achieved through convolutional computations and near-linear computational strategies in the Mamba architecture. Furthermore, a hierarchical query-based decoder is introduced, further enhancing model performance and efficiency. Extensive experimental results demonstrate that the proposed method achieves superior performance while maintaining low latency, which is crucial for practical real-time autonomous driving systems.

## 1 Introduction

Motion forecasting is a crucial component of autonomous driving systems, playing an important role in ensuring the safety of both drivers and pedestrians. It predicts agents' future trajectories based on their surrounding environment, which includes both dynamic surrounding agents and static map information. Given the inherent uncertainty in future behaviors, multiple plausible trajectories will be predicted to account for this ambiguity. Additionally, as autonomous driving is a resource-constrained system, efficiency is a key consideration for practical and real-time deployment.

In motion forecasting, the map can provide strong prior knowledge for predicting future trajectories. For example, the vehicles need to follow lanes. Depending on how the map information is represented, prior methods can be broadly classified into rasterized-based and vectorized-based approaches. Rasterized-based methods (Cui et al., 2019; Hong et al., 2019; Luo et al., 2018; Casas et al., 2018) represent the map as a 2D rasterized image, typically processed using a convolutional neural network (CNN) architecture. However, these approaches are often computationally heavy and inefficient for motion forecasting tasks. In contrast, vectorized-based methods (Gao et al., 2020; Liang et al., 2020b; Gao et al., 2020; Gu et al., 2021b; Zhou et al., 2022) process vectorized maps, which is a compact map representation and only compress lane information from high-definition (HD) maps. These methods commonly utilize graph convolutional networks (GCNs)(Liang et al., 2020a; Gao et al., 2020; Gu et al., 2021b; Zeng et al., 2021; Zhao et al., 2021) or transformer architectures (Gao et al., 2020; Liang et al., 2020b; Liu et al., 2021; Wang et al., 2022; Zhou et al., 2022) to process vectorized maps. Due to its strong ability to model spatial and temporal information, the transformer architecture has recently become the dominant approach in this domain.

Although transformer architecture has achieved significant success in the motion forecasting domain, it still faces several limitations. Autonomous driving is a resource-constrained system and needs to operate in a real-time environment, which demands highly efficient motion forecasting methods with minimal latency. However, the attention mechanism in transformers has quadratic complexity with respect to the input sequence length, making them computationally expensive. This issue becomes particularly pronounced in scenarios involving a large number of agents or lane segments, leading to increased latency. Moreover, transformers require substantial memory due to their multi-head attention mechanism and a large number of parameters, further complicating their deployment in resource-limited systems. Another key limitation is the lack of inductive biases. While positional encodings are often added to account for temporal dependencies or spatial relationships, they may not be as effective or intuitive as the inductive biases naturally embedded in convolutional architectures. This can potentially limit the transformer's performance in motion forecasting tasks, which are inherently spatially and temporally sensitive.

Recently, Mamba(Gu & Dao, 2023) was proposed as a more advanced foundation model, which has demonstrated superior efficiency and accuracy in various downstream tasks(Zhu et al., 2024b; Wang et al., 2024b). It originates from the classic state space models (SSMs) (Kalman, 1960) and excels in managing long sequences which is attributed to the implementation of convolutional computations and near-linear computational strategies (Gu et al., 2021a). Adapting selective state space modules for motion forecasting tasks presents notable challenges, primarily due to the lack of specialized design in SSMs for modeling spatial interaction. To address these challenges, we have carefully developed a motion forecasting architecture utilizing SSMs, named Multi-Stage State Space Model (MS$^3$M), specifically tailored to manage the complex spatial-temporal interactions within a scene, while optimizing computational efficiency with near-linear time complexity. In MS$^3$M the input data is converted into multiple tokens, each corresponding to a trajectory or lane segment in the scene. In this process, a Temporal Mamba Model (TMM) is employed to capture fine-grained temporal information. Unlike transformer architectures, where positional encodings are required, temporal dependencies in the Mamba Model are naturally encoded through its scanning operation. A stack of Single-Stage State Space Models (S$^4$Ms) is applied to these tokens to gradually model their spatial interactions. Within each S$^4$M, a spatial anchor is predicted, and the sequence of tokens is scanned based on their distance to this anchor. This process ensures that the model learns a spatial bias (anchor point), which is injected into the tokens through the scanning operation in the Mamba model. By injecting temporal and spatial inductive biases through Mamba's state-space model structure, the model's capacity is significantly improved. Finally, a hierarchical query-based decoder processes output tokens with different types sequentially, gradually aggregating information from them before decoding the future trajectories and their corresponding confidence scores.

MS$^3$M strikes an exceptional trade-off between accuracy and efficiency by adapting selective state space modules to effectively model both spatial and temporal information, making it well-suited for practical autonomous driving systems. Our contributions can be summarized as:

- We propose the Multi-Stage State Space Model (MS$^3$M), a pioneering multi-stage architecture that integrates a selective scanning mechanism into motion forecasting tasks. MS$^3$M achieves superior performance while significantly reducing model size and latency, making it more efficient for real-time autonomous driving systems.

- The Multi-Stage State Space Model (MS$^3$M) incorporates a Temporal Mamba Model to capture fine-grained temporal information and a Spatial Mamba Model to model spatial interactions. By injecting inductive biases of temporal and spatial dependency through Mamba's state-space model structure, the model's capacity is significantly improved.

- We propose a hierarchical query-based decoder, which further enhances model performance and efficiency by processing scene information in a structured and sequential manner.

## 2 RELATED WORK

### 2.1 MOTION FORECASTING

Motion Forecasting is a fundamental task in autonomous driving system, which predicts future trajectories according to current scenario. For accurate motion forecasting, two types of information

are usually required, spatial relationships to surrounding agents, like vehicles and pedestrians, at each timestep and temporal relationship for each agent across different timestep. To model spatial relationships , some previous works (Cui et al., 2019; Hong et al., 2019; Luo et al., 2018; Casas et al., 2018) represent the whole scene as a rasterized image and apply convolution neural network (CNN) on it, which may lose fine-grained scene details. As an comparison, vectorized representation attracts more attention as it can compress necessary information for autonomous driving. And graph neural networks (GNNs) (Liang et al., 2020a; Gao et al., 2020; Gu et al., 2021b; Zeng et al., 2021; Zhao et al., 2021) are usually utilized to process them. As for temporal relationship, recurrent neural networks (RNNs) (Mercat et al., 2020; Gupta et al., 2018; Alahi et al., 2016; Salzmann et al., 2020; Park et al., 2020) takes dominant position due to its excellent sequential data process ability. And some further works (Tang & Salakhutdinov, 2019; Djuric et al., 2020; Gilles et al., 2021; Rhinehart et al., 2019; Park et al., 2020) elaborate it with CNN for spatial-temporal trajectory prediction. Recently, the transformer architecture has gained significant attention due to its superior capability in modeling long-term dependencies. Due to its global perception ability, some recent motion forecasting work also utilize it for spatial relationship modeling (Gao et al., 2020; Liang et al., 2020b; Liu et al., 2021; Wang et al., 2022; Zhou et al., 2022). However, the standard transformer architecture (Vaswani, 2017) scales quadratically with the sequence length, making it inefficient when dealing with long sequences. Additionally, while transformers have a global receptive field, they do not inherently model temporal and spatial dependencies, relying instead on positional encodings, which can be suboptimal for motion forecasting task. To address these limitations, we introduce the State Space Model (SSM) (Gu & Dao, 2023), which offers linear computational complexity while maintaining a global receptive field like the transformer. Furthermore, it can explicitly model temporal dependencies, which is important for motion forecasting task. In this work, we propose a purely SSM-based motion forecasting model to overcome previous limitations.

## 2.2 State Space Models

State space models (SSMs) are fundamental tools for modeling dynamic systems, using a series of hidden variables to represent the system's evolution over time. Due to their ability to represent the recurrent process with latent states, SSMs are widely used in applications requiring the modeling of temporal dynamics, such as reinforcement learning (Hafner et al., 2020) and linear dynamical systems (Hespanha, 2018). While SSMs have broad applicability, they require significant computational and memory resources when modeling long-range dependencies. The following work (Gu et al., 2022) introduced the Structured State Space Sequence model (S4), which improves computational efficiency through parameterization techniques. Taking it a step further, (Fu et al., 2022) propose a novel SSM layer H3 based on S4 to narrow the gap between attention mechanism and SSMs in language modeling, optimizing both modeling capabilities and hardware efficiency. Inspired by the recently introduced Gated Attention Unit (Hua et al., 2022), the recent work(Mehta et al., 2023) proposes a layer named Gated State Space (GSS) to enhance the effectiveness of S4. Recently, Mamba (Gu & Dao, 2023) has gained increasing attention for its superior performance, achieved by introducing an input selection mechanism and a hardware-aware parallel algorithm. The input selection mechanism enables the model to selectively process data, reducing unnecessary computation on irrelevant parts of the sequence. With its linear complexity capabilities, Mamba has provided significant advantages in both natural language processing (Wang et al., 2024a; Liu et al., 2024; Zeng et al., 2024) and computer vision (Zhu et al., 2024a; Guo et al., 2024; Li et al., 2024; Liang et al., 2024; Lu et al., 2024). Despite these advancements, the use of a Mamba-based backbone in motion forecasting remains unexplored. In this work, we propose a Mamba-based solution to address this gap, achieving superior performance with significantly better efficiency.

## 3 Methodology

In this section, we outline the proposed Multi-Stage State Space Model (MS$^3$M), designed for motion forecasting under autonomous driving scenarios. Initially, we give a brief introduction to some related concepts, including motion forecasting task definition and the Selective State Space Model (Mamba) (Gu & Dao, 2023). Following this, we detail the proposed architecture that utilizes the Selective State Space Model (Mamba) to facilitate motion forecasting accuracy and efficiency.

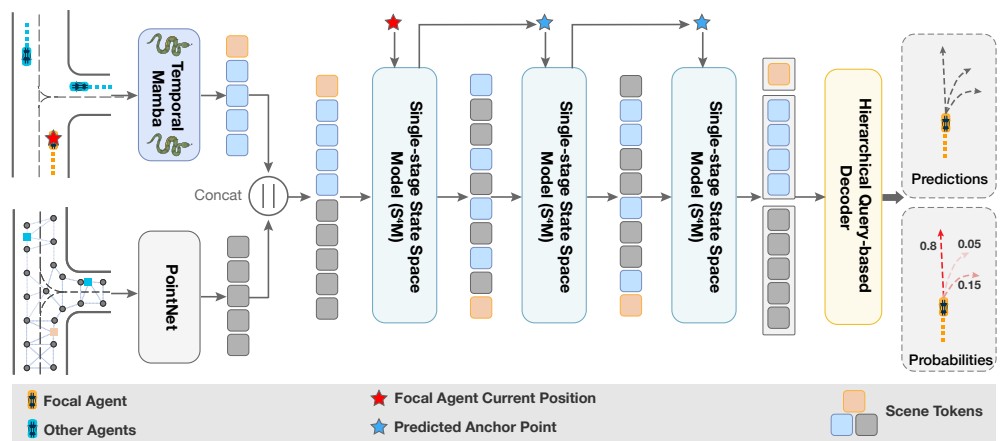

Figure 1: An overview of the proposed Multi-Stage State Space Model ($\text{M}S^3\text{M}$).

## 3.1 PRELIMINARIES

### 3.1.1 MOTION FORECASTING

Motion Forecasting in autonomous driving scenarios is usually defined as forecasting the future trajectory of a focal agent according to the current scenario. Because there usually exist multiple plausible future trajectories, the model is required to predict $K$ potential future trajectories and their corresponding probability score. This can be formulated as:

$$(\hat{\mathcal{T}}_k^f, \hat{s}_k)_{1:K} = \mathbf{Model}(\mathcal{T}_{0:N}^h, \mathcal{L}). \tag{1}$$

where $\hat{s}_k$ is the probability score for $k-th$ predicted trajectory. $\hat{\mathcal{T}}_{1:K}^f$ are predicted future trajectories:

$$\hat{\mathcal{T}}_{1:K}^f = \{\hat{x}_t : t \in \{1, \ldots, T_f\}\}_{1:K} \tag{2}$$

where $\hat{x}_t$ is the predicted 2D position at timestamp $t$. The model will receive the historical trajectories for both focal agent $\mathcal{T}_0^h$ and surrounding agents $\mathcal{T}_{1:N}^h$:

$$\mathcal{T}_{0:N_a}^h = \{x_t : t \in \{-T_h + 1, \ldots, 0\}\}_{0:N_a}. \tag{3}$$

where $t = 0$ represents the current timestamp and $x_t$ denotes the 2D position at timestamp $t$, our approach only consider the closest $N_a$ vehicles at timestamp $t = 0$. If there are less than $N_a$ surrounding agents, we mask the empty entries in $\mathcal{T}_{1:N}^h$. Map information often provides valuable information. In this work, we adopt a vectorized representation (Gao et al., 2020), which includes surrounding lanes and can be denoted as:

$$\mathcal{L} = \{x_i : i \in \{1, N_{pt}\}\}_{1:N_l} \tag{4}$$

where each lane is represented by $N_{pt}$ uniformly sampled 2D points from its centerline. We take only the $N_l$ closest lanes. Long lanes are split into multiple segments to ensure consistent distances between the sampled points, while for shorter lanes, missing points are masked.

Finally, we define the ground-truth future trajectory of the focal agent as:

$$\mathcal{T}^f = \{x_t : t \in \{1, \ldots, T_f\}\}, \tag{5}$$

### 3.1.2 SELECTIVE STATE SPACE MODEL

SSMs, notably through the innovations brought by structured state space sequence models (S4) and Mamba, have excelled in processing long sequences. These models transform a 1-D function or sequence, $x(t) \in \mathbb{R}$, into an output $y(t) \in \mathbb{R}$ through a hidden state $h(t) \in \mathbb{R}^N$. The evolution of the system is governed by $A \in \mathbb{R}^{N \times N}$, while $B \in \mathbb{R}^{N \times 1}$ and $C \in \mathbb{R}^{1 \times N}$ serve as the input and output projection matrices, respectively.

The discretized system can then be represented as follows, incorporating a step size $\Delta$:

$$h_t = Ah_{t-1} + Bt, \tag{6}$$

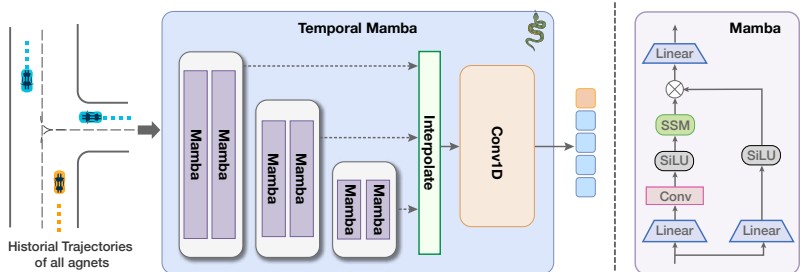

Figure 2: An overview of Temporal Mamba Module (TMM).

$$y_t = Ch_t. \tag{7}$$

This adaptation enables the computation of the output via global convolution, utilizing a structured convolutional kernel $K$ that spans the entire length $M$ of the input sequence $x$.:

$$K = (CB, CAB, \dots, CA^{M-1}B), \tag{8}$$

$$y = x * K. \tag{9}$$

Selective models like Mamba incorporate time-varying parameters, moving away from the linear time invariance (LTI) assumption and adding complexity to parallel computation. Nonetheless, hardware-aware optimizations, such as associative scans, have been introduced to mitigate these challenges, underscoring the continued advancement and application of SSMs in capturing complex temporal dynamics.

## 3.2 MULTI-STAGE STATE SPACE MODEL (MS$^3$M)

The architecture of the proposed Multi-Stage State Space Model (MS$^3$M) is illustrated in Figure 1 which fully utilizes the inherent long-sequence modeling capacity of the Mamba model and adapts it for spatial and temporal information modeling. In MS$^3$M, the scene encoder first converts each scene element, such as a trajectory or lane segment, into a separate token, where a Temporal Mamba Model (TMM) is utilized to capture fine-grained temporal information (Section. 3.2.1). The output tokens are then fed into a stack of Single-Stage State Space Models (S$^4$M) to model spatial interactions gradually. Within S$^4$M, the Spatial Mamba Model (SMM) learns a spatial inductive bias (anchor point) and injects it into the tokens through a scanning operation in the Mamba Model (Section. 3.2.2). Finally, the proposed hierarchical query-based decoder (Section. 3.2.3) will output future trajectories and their corresponding confidences by aggregating information from scene tokens in a structured and sequential manner.

### 3.2.1 SCENE ENCODER

Scene Encoder will convert each scene element, agent trajectory or lane segment, into a token separately to capture their inherent information. In the proposed scene encoder, agent trajectory and lane segment are processed differently.

We design a Temporal Mamba Module (TMM) to process each agent trajectory $\mathcal{T}_i^h$ which is shown in Figure 2. The Temporal Mamba Module utilizes a Feature Pyramid Network (FPN) architecture to capture fine-grained temporal information at different scales. It comprises multiple stages with decreasing resolution, with each stage consisting of several Mamba models. With the scanning operation in the Mamba model, the temporal dependency is directly captured. Finally, the multi-scale temporal features will be fused in the end. Additionally, we add a semantic class embedding $Cls_i^A$ to inject semantic information. This can be formulated as:

$$\mathcal{ST}_i^A = \mathbf{TMM}(\mathcal{T}_i^h) + Cls_i^A, \tag{10}$$

where $Cls_i^A$ denotes the type information of agents such as vehicles or pedestrians.

Unlike trajectories, where temporal connections are crucial, the spatial relationships within lane segments are of greater importance. Considering that there are typically many more lane segments

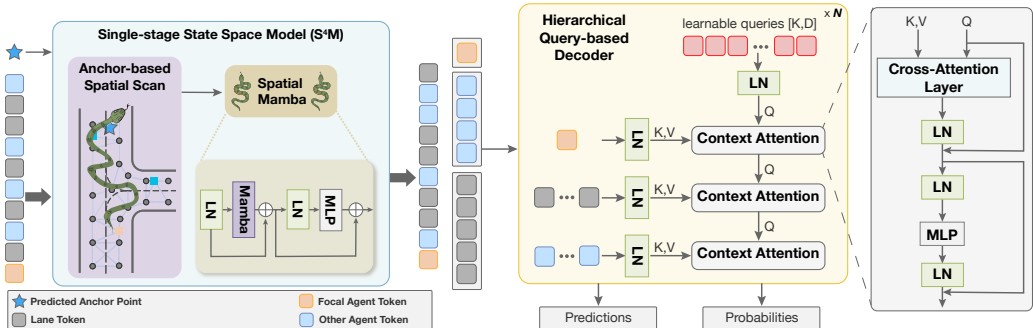

Figure 3: An overview of Single-stage State Space Model ($S^4$M)

Figure 4: Hierarchical Query-based Decoder

than agents, we employ a lightweight mini-PointNet (Qi et al., 2017) to learn the lane embeddings, prioritizing efficiency in the process. :

$$\mathcal{ST}_i^L = \textbf{MiniPointNet}(\mathcal{L}_i) + Cls_i^L, \tag{11}$$

Similarly, $Cls_i^L$ represents lane types and is initialized to a learnable embedding.

Thus the scene encoder will convert the input scene representation into multiple scene tokens $\mathcal{ST}$:

$$\mathcal{ST} = (\mathcal{ST}^A || \mathcal{ST}^L) \tag{12}$$

where $||$ denotes the concatenation operator $\mathcal{ST}^A = \{\mathcal{ST}_i^A : i \in 0, ..., N_a\}$ and $\mathcal{ST}^L = \{\mathcal{ST}_i^L : i \in 0, ..., N_l\}$ are tokens correspond to agents and lanes separately . With this design, we strike a balance between model efficiency and performance by leveraging the Mamba model and allocating resources accordingly.

### 3.2.2 MULTI-STAGE ARCHITECTURE

A multi-stage architecture which is a stack of Single-Stage State Space Model ($S^4$M) consisting of the Mamba model will be applied to model spatial interaction among scene tokens $\mathcal{ST}$ gradually.

**Single-Stage State Space Model ($S^4$M)**

We first introduce the Single-Stage State Space Model ($S^4$M), shown in Figure. 3, which is used to model spatial interactions among scene tokens. At its core is the Spatial Mamba Model (SMM), which learns a spatial inductive bias (anchor point) and injects it into scene tokens through an Anchor-based Spatial Scan (ASS) mechanism.

The Anchor-based Spatial Scan (ASS) mechanism is used to organize scene tokens $\mathcal{ST}$ into ordered scene tokens $\mathcal{OST}$. The Mamba Model adopts a scanning mechanism, where tokens are processed sequentially, to efficiently handle long sequences. In this process, the order of tokens is crucial, but this information is missing in the raw scene tokens $\mathcal{ST}$. To address this, the Anchor-based Spatial Scan (ASS) reorders scene tokens $\mathcal{ST}$ based on a predicted anchor point $ap$. Finally, the Spatial Mamba Model (SMM) scans the scene tokens according to this new order to model the spatial interactions among them effectively.

For $n$-th Single-Stage State Space Model ($S^4M_n$), it will receive scene tokens $\mathcal{ST}_{n-1}$ and a predicted anchor point $ap_{n-1}$ produced from the last stage $(n-1)$-th. The output will be new scene tokens $\mathcal{ST}_n$ and $K$ predicted anchor points $\hat{AP}_n = \{ap_{n,1}, ...ap_{n,K}\}$ and their corresponding scores $\hat{S}_n = \{s_{n,1}, ..., s_{n,K}\}$ which can be formulated:

$$\mathcal{ST}_n, \hat{AP}_n, \hat{S}_n = Stage_i(\mathcal{ST}_{n-1}, ap_{n-1}) \tag{13}$$

where for the first stage, $ap_{n-1}$ equals to the current position of the focal agent ($t = 0$).In the following stages, the anchor point $ap_{n-1}$ is directly predicted from the previous stage. Since multiple plausible predictions may exist, $S^4$M is designed to output $K$ anchor points along with their corresponding confidence scores. The anchor point with the highest confidence score is then selected to capture the most likely outcome. This can be formulated as:

$$ap_{n-1} = \hat{AP}_{n-1}[idx], \qquad idx = argmax\hat{S}_{n-1} \tag{14}$$

Then the input scene tokens $\mathcal{ST}_n$ will be reordered according to the Euclidean distance $L_2$ from each scene token to the predicted anchor point $ap_{n-1}$:

$$\mathcal{OST}_{n-1} = \{\mathcal{ST}_{n-1}^{(1)}, ..., \mathcal{ST}_{n-1}^{(N+N_l)}\} \quad , \quad d(ap_{n-1}, \mathcal{ST}_{n-1}^{(1)}) \leq \cdots \leq d(ap_{n-1}, \mathcal{ST}_{n-1}^{(N+N_l)}) \tag{15}$$

where $d(ap_{n-1}, \mathcal{ST}_{n-1}^{(i)}) = ||POS(\mathcal{ST}_{n-1}^{(i)}) - ap_{n-1}||$ denotes the Euclidean distance between each scene token $\mathcal{ST}_{n-1}^{(i)}$ and anchor point $ap_{n-1}$. $POS(\mathcal{ST}_{n-1}^{(i)})$ denotes the current position if the scene token is an agent token. Otherwise, it denotes the closest point at the lane to anchor point $ap_{n-1}$. One important detail to note is that the scene token corresponding to the focal agent is always placed at the end of the sequence. This ensures that it can aggregate information from all the preceding scene tokens. The Spatial Mamba Model (SMM) will scan the ordered scene tokens $\mathcal{OST}_{n-1}$ to update their spatial relationships. It consists of layer normalization, Mamba model, layer normalization, and multilayer perceptron (MLP) layer sequentially. Additionally, the residual linked will be added accordingly. The detailed architecture is shown in Figure. 3.

Compared to previous transformer-based methods, which typically utilize attention mechanisms for information aggregation and append positional encodings to represent spatial relationships, the proposed S$^4$M directly learns a spatial inductive bias (anchor point) and injects it into the scene tokens through an anchor-based spatial scan mechanism. With the implementation of convolutional computations and near-linear computational strategies in the Mamba architecture, S$^4$M is also significantly more efficient.

Finally, the output scene token corresponding to the focal agent will output $K$ anchor points and confidence scores for the following stage by two multilayer perceptron (MLP) layers:

$$\hat{AP}_n = MLP_{ap}^n(\mathcal{ST}_n^{A,0}) \tag{16}$$

$$\hat{S}_n = MLP_{score}^n(\mathcal{ST}_n^{A,0}) \tag{17}$$

We observe the endpoint of the future trajectory of the focal agent usually contributes to the final performance. Therefore, we enforce the best-predicted anchor point at each stage (Equation. 14) to align with this endpoint which will be shown later.

**Multi-Stage State Space Model (MS$^3$M)**

Stacking multiple predictors sequentially has demonstrated significant improvements in various tasks, such as human pose estimation (Xu & Takano, 2021; Wei et al., 2016). Inspired by these works, we sequentially stack several Single-Stage State Space Models (S$^4$M). In this multi-stage model, each stage processes the scene tokens $\mathcal{ST}$ along with an anchor point $ap$ provided by the previous stage. By gradually refining the anchor point, which influences the scan order in the Mamba model, the overall performance is progressively enhanced.

### 3.2.3 HIERARCHICAL QUERY-BASED DECODER

The DETR-like query-based decoder (Carion et al., 2020) is widely adopted for motion forecasting, where all scene tokens are treated equally and processed together. However, this approach is suboptimal as it neglects the inherent attributes of each token. For instance, the focal agent token typically contributes more significantly to the final performance compared to other tokens. To address this limitation, we propose a Hierarchical Query-based Decoder that treats scene tokens differently based on their semantic classes. Specifically, we introduce $K$ learnable queries, $Q \in \mathbb{R}^{K \times D}$, where each query is responsible for decoding one of the $K$ future trajectory modes. These mode queries are updated incrementally by sequentially feeding in scene tokens of different types. The detailed architecture is shown in Figure 4.

Finally, the focal agent token will be projected into physical space using two separate multilayer perceptron (MLP) layers, producing the predicted trajectories of the focal agent and the corresponding probability for each mode.

### 3.3 SUPERVISION

We apply different supervision after each stage of the model. For the final stage, we utilize the widely used smooth L1 loss for trajectory regression and cross-entropy loss for confidence classification.

| Method | b-minFDE$_6$ | minADE$_6$ | minFDE$_6$ | MR$_6$ | minADE$_1$ | minFDE$_1$ | MR$_1$ |
|---|---|---|---|---|---|---|---|
| GoRela(Cui et al., 2023) | 2.01 | 0.76 | 1.48 | 0.22 | 1.82 | 4.62 | 0.66 |
| THOMAS(Gilles et al., 2022) | 2.16 | 0.88 | 1.51 | 0.20 | 1.95 | 4.71 | 0.64 |
| MTR (Shi et al., 2022) | 1.98 | 0.73 | 1.44 | 0.15 | 1.74 | 4.39 | 0.58 |
| GANet (Wang et al., 2023) | 1.96 | 0.72 | 1.34 | 0.17 | 1.77 | 4.48 | 0.59 |
| QCNet (Zhou et al., 2023) | **1.91** | **0.65** | **1.29** | **0.16** | **1.69** | 4.30 | 0.59 |
| MS$^3$M (1 stage) | 2.10 | 0.75 | 1.48 | 0.20 | 1.89 | 4.72 | 0.64 |
| MS$^3$M (2 stages) | 2.02 | 0.72 | 1.39 | 0.17 | 1.74 | 4.35 | 0.61 |
| MS$^3$M (3 stages) | 2.08 | 0.74 | 1.43 | 0.18 | 1.70 | **4.20** | 0.60 |
| MS$^3$M (4 stags) | 2.10 | 0.75 | 1.45 | 0.18 | 1.72 | 4.23 | 0.60 |
| QML* (Su et al., 2022) | 1.95 | 0.69 | 1.39 | 0.19 | 1.84 | 4.98 | 0.62 |
| BANet* (Zhang et al., 2022) | 1.92 | 0.71 | 1.36 | 0.19 | 1.79 | 4.61 | 0.60 |
| QCNet * (Zhou et al., 2023) | **1.78** | **0.62** | **1.19** | **0.14** | **1.56** | **3.96** | **0.55** |
| MS$^3$M (Stage 1-4) * | 1.91 | 0.68 | 1.30 | 0.16 | 1.64 | 4.08 | 0.58 |

Table 1: Comparison of motion forecasting methods on Argoverse 2 test set. Baselines that are known to have used ensembling are marked with the symbol "*". For each metric, the best result is in **bold** and the second best result is underlined.

Additionally, we employ the winner-take-all strategy, which optimizes only the best prediction—i.e., the one with the minimal final prediction error compared to the ground truth. For all preceding stages, we enforce that the best-predicted anchor point, as described in Equation 14, aligns with the endpoint of the focal agent's future trajectory. Appropriate weights are assigned to different loss terms to balance their contributions effectively.

# 4 EXPERIMENTS

## 4.1 EXPERIMENT SETTING

**Dataset** We compare the proposed method to previous state-of-the-art methods on popular large-scale Argoverse2 (AV2) dataset. This dataset includes 199,908 sequences for training, 24,988 sequences for validation, and 24,984 sequences for testing. Each sequence is sampled at 10 Hz, with 5 seconds of historical data and a requirement to predict 6 seconds into the future (i.e., $T_h = 50$, $T_f = 60$).

**Evaluation Metrics** In line with Argoverse 2 official online benchmark metrics, we use the minimum Average Displacement Error (minADE$_K$), minimum Final Displacement Error (minFDE$_K$), Miss Rate (MR$_K$), and Brier-minimum Final Displacement Error (b-minFDE$_K$) for evaluation. These metrics permit models to predict up to $K$ trajectories per agent, with $K$ set to 1 and 6 for consistency with previous methods.

## 4.2 COMPARISON TO STATE-OF-THE-ART METHODS

We first compare the proposed Multi-Stage State Space Model (MS$^3$M) to the state-of-the-art methods on the Argoverse 2 online benchmark (test set), as shown in Table. 1. The results indicate that MS$^3$M achieves performance comparable to the current state-of-the-art method, QC-Net (Zhou et al., 2023), a pure transformer-based architecture. Even without ensembling, MS$^3$M demonstrates strong performance, ranking second-best across most metrics with different numbers of stages. This suggests that the performance of the individual stages complements each

| Method | Latency (ms) | Model size(M) |
|---|---|---|
| QCNet | 54.55±17.2 | 7.7M |
| MS$^3$M (1 stage) | 16.56±26.38 | 4.6M |
| MS$^3$M (2 stages) | 20.25±26.24 | 5.9M |
| MS$^3$M (3 stages) | 22.56±26.56 | 7.2M |
| MS$^3$M (4 stages) | 25.54±26.65 | 8.4M |

Table 2: Latency and Model size comparison. Even though QCNet (Zhou et al., 2023) reuses computations from previous observation windows, reducing latency by over 6× as indicated in (Zhou et al., 2023), MS$^3$M still achieves significantly lower latency, regardless of the number of stages.The experiment was conducted on a single NVIDIA RTX A5000.

other. For example, MS$^3$M with 2 stages performs better on metrics like minADE$_6$ and MR$_6$, while MS$^3$M with 3 stages excels in minADE$_1$ and minFDE$_1$. This motivated us to ensemble MS$^3$M with

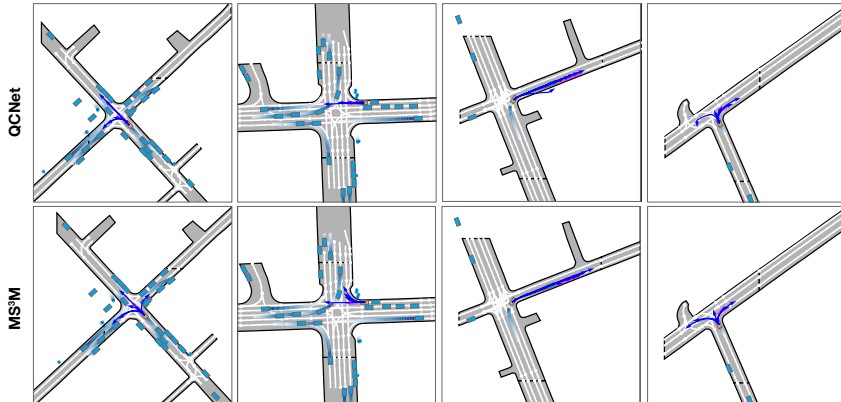

Figure 5: Qualitative results. We compare MS$^3$M to the state-of-the-art method, QCNet (Zhou et al., 2023). Blue arrows represent the predicted future trajectories (K=6), while the pink arrow denotes the ground truth future trajectory. The orange bounding box indicates the focal agent while the blue bounding boxes denote surrounding agents. The proposed method demonstrates the ability to produce diverse (Columns 1 and 2) yet accurate (Columns 3 and 4) predictions. In certain scenarios, QCNet generates implausible predictions (Columns 3 and 4), which are avoided by the proposed method. This highlights the strong spatial and temporal modeling capabilities of the proposed approach.

different stages, leading to a clear improvement in performance and enabling MS$^3$M to surpass most previous methods.

The superior performance of QCNet (Zhou et al., 2023) comes at the cost of high latency, as shown in Table. 2. We observe that even though MS$^3$M with 4 stages has more parameters compared to QCNet (Zhou et al., 2023), it still reduces latency by more than half. It is important to note that QC-Net (Zhou et al., 2023) reuses computations from previous observation windows, reducing latency by over 6×, as indicated in (Zhou et al., 2023). Despite not employing such optimizations, MS$^3$M still achieves significantly lower latency. Furthermore, reducing the number of stages in MS$^3$M further widens the latency gap. Currently, MS$^3$M with 2 or 3 stages achieves the best performance, outperforming QCNet (Zhou et al., 2023) in both model size and latency by a large margin.

Finally, we present some qualitative results in Figure. 5, where we observe that MS$^3$M produces future trajectories that are both as accurate and diverse as those generated by QCNet. In some scenarios, the trajectories predicted by MS$^3$M even appear more reasonable, further highlighting its effectiveness.

### 4.3 ABLATION STUDY

Next, we conduct some ablation studies on the Argoverse 2 validation set to demonstrate the effectiveness of our designs. The experimental results are presented in Table. 3.

**Decoder Design** We first explore different designs for the query-based decoder. For the traditional query-based decoder ("non-Hier" in Table. 3) used in motion forecasting, which feeds all tokens as key and value into the attention layer simultaneously, we ensure a fair comparison by using the same number of attention layers as the proposed hierarchical query-based decoder ("Hier" in Table. 3). We observe that feeding scene tokens in a structured and sequential manner based on their different types leads to overall better performance. This improvement is primarily due to the reduction of ambiguity within the input tokens for each attention layer. Additionally, by processing 3× fewer tokens as key and value, the model's efficiency is further enhanced, reducing computational complexity and improving runtime performance without sacrificing accuracy.

**Intermediate Supervision Choice** We also investigate the influence of added supervision at intermediate stages. A straightforward approach is to apply the same supervision across all stages, meaning the supervision used in the final stage, as described in Section 3.3, is also applied to all preceding stages. However, we found that this strategy ("best FDE" in Table. 3) does not perform as well as our current solution ("best prob" in Table. 3). We assume that selecting the trajectory

| Method | | minADE$_6$ | minFDE$_6$ | MR$_6$ | minADE$_1$ | minFDE$_1$ |
|---|---|---|---|---|---|---|
| Decoder | non-Hier | 0.735 | 1.435 | 0.177 | 1.692 | 4.200 |
| | Hier | 0.732 | 1.429 | 0.177 | 1.687 | 4.179 |
| Loss type | best fde | 0.732 | 1.432 | 0.179 | 1.700 | 4.234 |
| | best prob | 0.732 | 1.429 | 0.177 | 1.687 | 4.179 |
| Stage Effect | Deep S$^4$M | 0.746 | 1.471 | 0.199 | 1.853 | 4.634 |
| | MS$^3$M | 0.732 | 1.429 | 0.177 | 1.687 | 4.179 |

Table 3: Ablation Study.

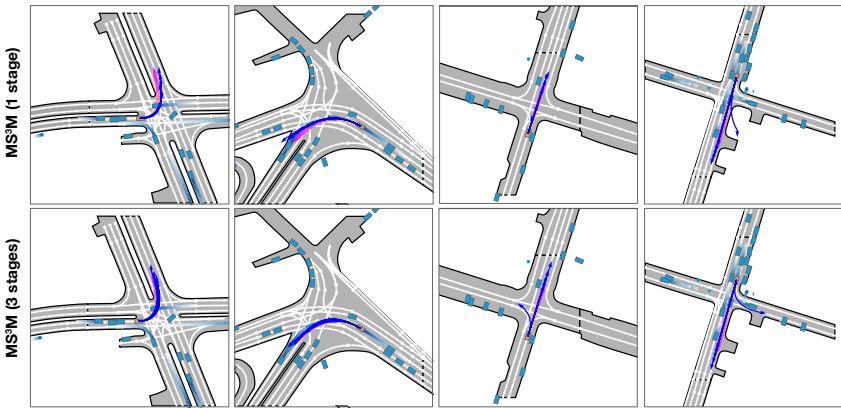

Figure 6: Qualitative results for the different number of stages. We observe that increasing the number of stages leads to more accurate predictions (Columns 1 and 2), whereas predictions from a single stage are often less precise, frequently deviating from the ground truth or even entering non-driving areas. Furthermore, increasing the number of stages also results in more diverse predictions, as shown in Columns 3 and 4.

with the max probability in the intermediate stages allows the model to focus on the most likely and plausible outcomes, ensuring it explores realistic scenarios without prematurely narrowing its focus on less probable trajectories.

**Deep Single-Stage Model** In the previous section, we demonstrated that our multi-stage architecture outperforms a single-stage model. However, that comparison alone doesn't clarify whether the improvement stems from the multi-stage design itself or simply from the increase in parameters as more stages are added. To ensure a fair comparison, we train a single-stage model with the same number of parameters as the multi-stage version. As shown in Table. 3, our multi-stage architecture significantly outperforms the single-stage counterpart ("Deep S$^4$M" in Table. 3), highlighting the effectiveness of the proposed architecture in enhancing prediction quality.

Finally, we present some qualitative results to illustrate the impact of varying the number of stages, shown in Figure. 6. Please refer to the supplementary material for additional ablation studies and qualitative results.

## 5 CONCLUSION

In this paper, we propose a Multi-Stage State Space Model (MS$^3$M) for motion forecasting in autonomous driving scenarios. Compared to previous dominant transformer-based methods, the proposed approach strikes an exceptional balance between accuracy and efficiency by leveraging Mamba model for both spatial and temporal information modeling. The Temporal Mamba Model effectively captures fine-grained temporal information, while spatial interactions among scene elements are modeled through the Single-Stage State Space Model (S$^4$M). Within S$^4$M, a spatial inductive bias (anchor point) is learned and injected into scene tokens via Mamba's state-space model structure. Additionally, a hierarchical query-based decoder is introduced, further enhancing both model performance and efficiency. Extensive experimental results demonstrate that the proposed approach achieves superior performance while maintaining high computational efficiency, making it well-suited for practical real-time autonomous driving systems.

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
