# SUPPLEMENT MATERIAL FOR "MS³M: MULTI-STAGE STATE SPACE MODEL FOR MOTION FORECASTING"

## A    IMPLEMENTATION DETAILS

In all experiments, we utilized the AdamW optimizer with a cosine learning rate scheduler. The training process consisted of 70 epochs, with the first 10 epochs serving as the warm-up phase. During this period, the learning rate gradually increased to its base value of 1e-3. After the warm-up, the learning rate decayed following a cosine schedule. A weight decay of 1e-4 was applied to regularize the model and prevent overfitting. All experiments were conducted with a batch size of 32. Our Single-Stage State Space Model ($S^4M$) includes 4 Spatial Mamba layers. The supervision weights in our work are fixed, with a value of 0.25 in stage 1 and 0.5 in stage 2.

## B    EFFECT OF SUPERVISION WEIGHTS ON INTERMEDIATE STAGES

As mentioned in Section 3.3, appropriate weights are assigned to different loss terms to balance their contributions effectively. We conduct a detailed analysis of the impact of these weights, and the results are presented in Table. 1.We experimented with both fixed loss weights and linear decay weights, and found that using fixed weights of 0.25 for stage 1 and 0.5 for stage 2 achieves the best performance.

Table 1: The impact of supervision weights on intermediates stages.

| Method | Stage1 | Stage2 | minADE6 | minFDE6 | MR6 | minADE1 | minFDE1 |
|---|---|---|---|---|---|---|---|
| | 1.0 | 1.0 | 0.741 | 1.444 | 0.180 | 1.698 | 4.2 |
| | 0.5 | 0.5 | 0.734 | 1.425 | 0.177 | 1.697 | 4.212 |
| fix weight | 0.5 | 0.75 | 0.735 | 1.427 | 0.178 | 1.70 | 4.213 |
| | 0.25 | 0.5 | 0.732 | 1.429 | 0.177 | 1.687 | 4.179 |
| | 0.25 | 0.25 | 0.727 | 1.423 | 0.179 | 1.694 | 4.214 |
| | [1.0, 0.5] | [1.0, 0.5] | 0.740 | 1.444 | 0.180 | 1.694 | 4.2 |
| linear decay weight | [1.0, 0.0] | [1.0, 0.0] | 0.735 | 1.423 | 0.177 | 1.708 | 4.239 |
| | [0.5, 0.0] | [0.5, 0.0] | 0.722 | 1.412 | 0.178 | 1.712 | 4.274 |

## C    EFFECT OF LAYER NUMBER IN EACH STAGE

We investigate the effect of varying the number of Spatial Mamba layers in the Single-Stage State Space Model ($S^4M$), and the results are presented in Table. 2. We can observe when 4 Spatial Mamba layers are adopted inside each $S^4M$, the proposed method achieves the best performance.

Table 2: The impact of the number of Spatial Mamba layers in the Single-stage State Space Model($S^4M$).

| Layer nums | minADE6 | minFDE6 | MR6 | minADE1 | minFDE1 |
|---|---|---|---|---|---|
| 1 | 0.725 | 1.409 | 0.181 | 1.721 | 4.308 |
| 2 | 0.728 | 1.423 | 0.181 | 1.716 | 4.273 |
| 4 | 0.732 | 1.429 | 0.177 | 1.687 | 4.179 |
| 8 | 0.746 | 1.462 | 0.182 | 1.684 | 4.181 |

## D    EFFECT OF SPATIAL BIAS

We conduct a toy experiment to investigate the impact of spatial bias (anchor point). In this experiment, we directly inject the endpoint of the focal agent's future trajectory into a Single-Stage State Space Model ($S^4M$) to observe its influence. Additionally, we introduce some disturbance to the anchor point to evaluate its effect on performance. For comparison, we also present the performance of the proposed $MS^3M$. The experimental results are shown in Figure. 1. We can observe the spatial bias (predicted anchor point) should be as close to the endpoint of the focal agent's future trajectory as possible.

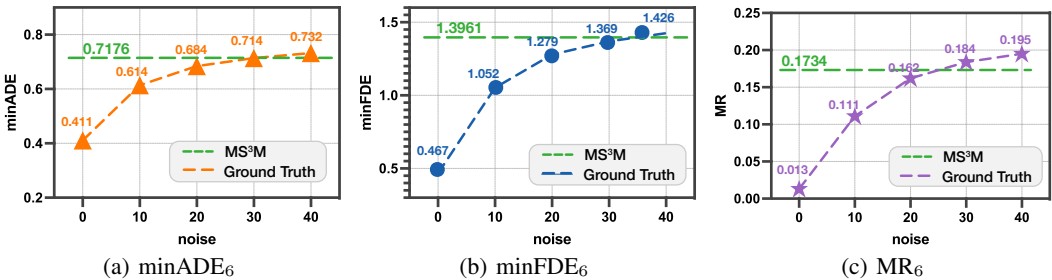

|   |   |   |
|---|---|---|
| (a) minADE$_6$ | (b) minFDE$_6$ | (c) MR$_6$ |

Figure 1: Effect of Spatial Bias: We introduce a uniform disturbance with the mean value indicated on the x-axis. The results show that the model's performance is excellent when no disturbance is added. As the disturbance value increases, performance gradually declines. This suggests that if the model can learn the spatial bias accurately—meaning the predicted anchor is as close to the endpoint of the focal agent's future trajectory as possible—it will predict future trajectories more accurately.

## E    QUALITATIVE VISUALIZATION

We present additional qualitative visualizations to demonstrate the impact of varying the number of stages, as shown in Figure. 2.

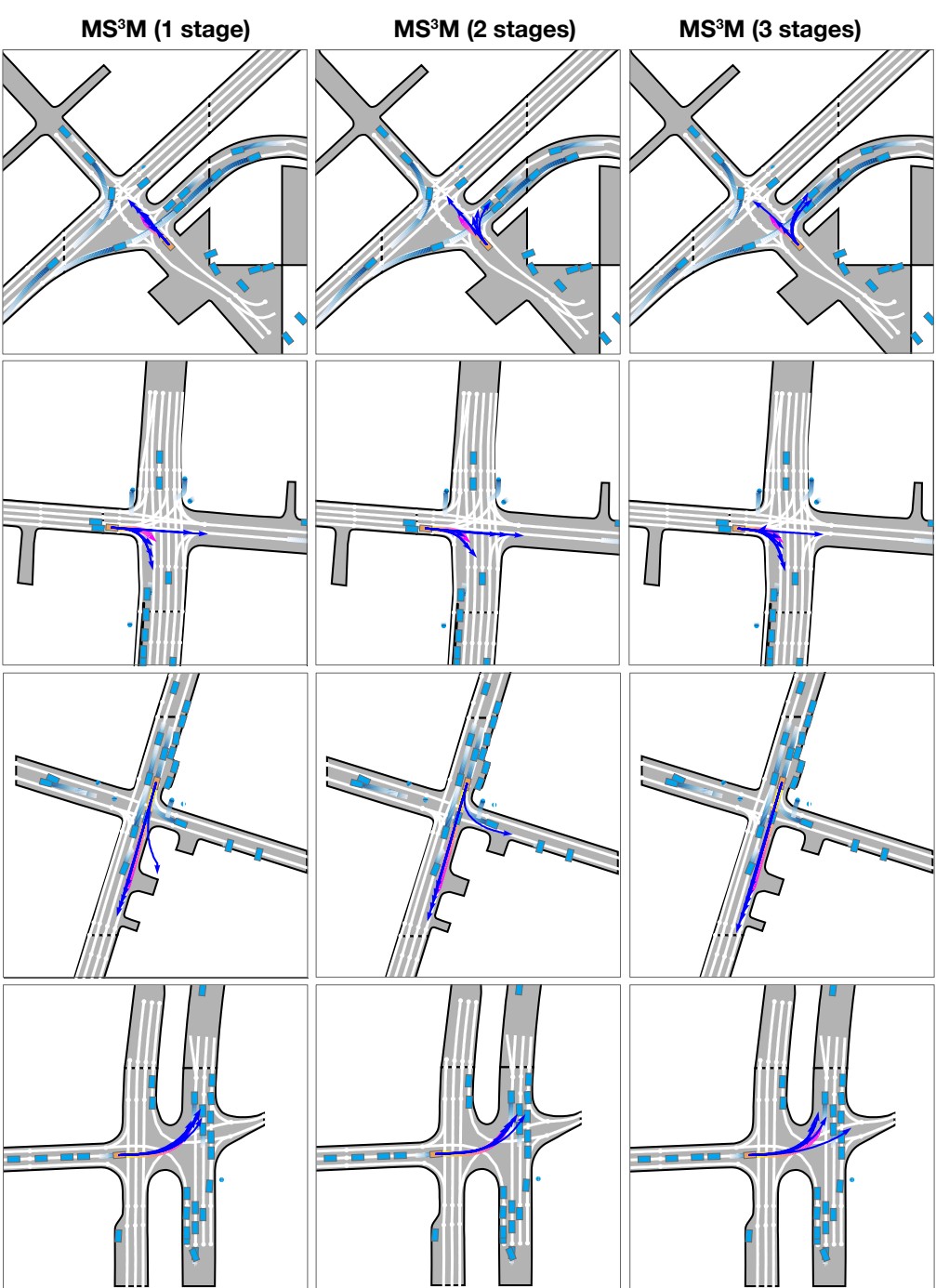

Figure 2: More visualization results and comparisons on Argoverse 2 validation set. The predicted future trajectories (K=6) are in blue and the ground truth future trajectories are in gradient pink. The orange bounding box indicates the focal agent while the blue bounding boxes denote surrounding agents. It can be seen that when increase in the number of stages leads to a higher accuracy in the predicted trajectories. The model can retain the diversity of generated modes while effectively eliminating unrealistic predictions.