# OpenReview forum: "MS$^3$M: Multi-Stage State Space Model for Motion Forecasting"
_ICLR.cc/2025/Conference — Submitted to ICLR 2025_

### Official Review · Reviewer_Dxg8 · 2024-10-31

**Soundness:** 3
**Presentation:** 2
**Contribution:** 2
**Rating:** 3
**Confidence:** 5

**Summary:**

This paper proposes an motion prediction model based on the Mamba architecture, which consists of the Temporal Mamba Model and the Spatial Mamba Model that handle temporal and spatial dependencies respectively. By leveraging the Mamba architecture's near-linear computational complexity with respect to sequence length, the proposed method demonstrates efficiency advantages compared to QCNet, a transformer-based approach.

**Strengths:**

1. The paper is relatively well-organized and easy to read.
2. The paper introduces the concept of state space models into the scene encoding of motion prediction, which is a noteworthy innovation.

**Weaknesses:**

1. The authors mention in the contributions that 'By injecting inductive biases of temporal and spatial dependency through Mamba’s state-space model structure, the model’s capacity is significantly improved.' The authors should clarify for this improvement relative to which specific methods.
2. Based on the test results from Argoverse 2, the proposed method shows a significant performance gap in various accuracy metrics compared to Transformer-based methods like QCNet. Although MS$^3$M has the best performance on minFDE$_1$, this is achieved by excluding some more advanced methods such as SEPT and LOF. Therefore, the experiments do not support the claim in the paper that "Mamba was proposed as a more advanced foundation model".
3. The experimental results are not sufficiently convincing, as the authors only conducted experiments on the Argoverse 2. It is recommended to include experimental results on the Argoverse 1 or other datasets.
4. In the test for inference time, the paper presents some unusual data, such as '16.56±26.38.' I suggest that the authors need to explain why the variance for the MS$^3$M is so large.

**Questions:**

1. Among the compared methods, some SOTA approaches listed on the leaderboard, such as SEPT and LOF, are not included. Please clarify the criteria used for selecting the baselines.
2. In the scene encoding, the tokens corresponding to lane should have a natural permutation invariance property. Why do the authors assert that 'the order of tokens is crucial'? Additionally, what is the specific advantage of using a state-space model, a typical temporal model, to handle the spatial structural relationships in traffic scenes?

---

### Official Review · Reviewer_jjYs · 2024-11-01

**Soundness:** 2
**Presentation:** 3
**Contribution:** 3
**Rating:** 5
**Confidence:** 3

**Summary:**

This paper presents the Multi-Stage State Space Model (MS$^3$M), a new architecture that leverages recent advancements in state space models to improve motion forecasting in autonomous driving systems by effectively modeling spatial-temporal information. This approach aims to address the inefficiencies of traditional transformer models that suffer from quadratic complexity as the number of agents increases. The paper introduces the Temporal Mamba Model and Spatial Mamba Model to handle temporal information and spatial interactions, respectively. Additionally, a hierarchical query-based decoder further enhances both performance and efficiency by structuring scene processing. This paper includes experiments and ablation studies on the Argoverse 2 dataset to demonstrate the effectiveness of MS3M and the trade-off it achieves between latency and performance.

**Strengths:**

The paper provides a well-articulated introduction to the methods and thorough explanations of the experiments. It introduces the Mamba architecture to handle long sequences and complex spatial-temporal interactions in motion prediction, which represents an interesting and promising direction especially for lighter model with lower latency solution in autonomous driving.

**Weaknesses:**

1. It would be beneficial to show the model's performance on additional benchmarks, such as WOMD/NuScenes/Argoverse 1 to strengthen the conclusion that this approach achieves a good trade-off between efficiency and accuracy, given that the best performance of MS$^3$M is still behind the baseline model.
2. A recent paper [1] also demonstrates significant benefits from the Mamba architecture, and the paper [2] that utilizes LM also shows low latency with reasonable accuracy. It would be interesting to have a deeper analysis of the contributions to model size and latency among different architectures such as CNN/transformer/Mamba.
3. The caption of Table 3 L493 could be improved to explain the ablation details, making it more readable.

[1] Zhang, Bozhou, Nan Song, and Li Zhang. "Decoupling Motion Forecasting into Directional Intentions and Dynamic States." arXiv preprint arXiv:2410.05982 (2024).

[2] Seff, Ari, et al. "Motionlm: Multi-agent motion forecasting as language modeling." Proceedings of the IEEE/CVF International Conference on Computer Vision. 2023

**Questions:**

Paper [1] also evaluates the latency of QCNet on the Argoverse dataset (Table 2&8), and the latency is much higher. Could this be due to different settings?

[1] Zhou, Yang, et al. "SmartRefine: A Scenario-Adaptive Refinement Framework for Efficient Motion Prediction." Proceedings of the IEEE/CVF Conference on Computer Vision and Pattern Recognition. 2024.

---

### Official Review · Reviewer_g3Ds · 2024-11-02

**Soundness:** 2
**Presentation:** 2
**Contribution:** 3
**Rating:** 3
**Confidence:** 5

**Summary:**

For ego-vehicle motion forecasting, the authors leverage recent advances in state space models (SSMs) and propose the Multi-Stage State Space Model (MS3M). In MS3M, the Temporal Mamba Model is employed to capture fine-grained temporal information, while the Spatial Mamba Model handles spatial interactions. Furthermore, a hierarchical query-based decoder is introduced.

To demonstrate the effect of the proposed approach, the authors conducted experiments on the large-scale relevant Argoverse2 dataset.

**Strengths:**

1. The authors propose the Multi-Stage State Space Model (MS3M), a novel multi-stage architecture that integrates a selective scanning mechanism into motion forecasting task. It incorporates a Temporal Mamba Model to capture fine-grained temporal information and a Spatial Mamba Model to model spatial interactions.
2. The authors also propose an original hierarchical query-based decoder.
3. The authors conducted experiments with the relevant and sufficiently diverse Argoverse2 dataset.

**Weaknesses:**

1. A significant drawback of the article is the small number of experiments that would confirm the hypotheses on the quality and performance of the proposed approach compared to state-of-the-art approaches. In addition to Argoverse2 for the same task, there are benchmarks for motion forecasting based on Waymo Open Dataset, Lyft, Nuscenes, on which the quality of the proposed approach was not tested. Thus, I recommend the authors to conduct additional experimental validation of their approach on the mentioned datasets. This would raise important questions about the performance of the method and its generalizability to different data distributions.
2. The positive effect and applicability of the proposed approach is questionable. In Table 1, the authors themselves show that the developed MS3M approach is slightly inferior to the QCNet transformer approach, and in Table 2, the latency ranges of these methods significantly overlap taking into account the variance. Thus, the authors' thesis that the proposed method "achieves superior performance while significantly reducing model size and latency" is questionable. I suggest that the authors clarify or revise their claims about superior performance and reduced latency in light of the overlapping ranges.
3. The comparison of methods does not include approaches that appeared in 2024, such as UniTraj, JointMotion, etc. I recommend the authors to add an updated comparison to the paper and discuss what fundamental advantages MS3M may have over these recent methods.
4. The text of the paper requires the elimination of typos and inaccuracies, for example, some formulas lack punctuation marks at the end (commas or dots).

**Questions:**

1. Judging by the scheme of the proposed Multi-Stage State Space Model (MS3M), one of the input data is the trajectories of other agents, it would be interesting to see the dependence of the performance of the proposed method on the number of these agents.
2. How would this dependence compare with the result of transformer models, for example QCNet?

---

### Official Review · Reviewer_fbPc · 2024-11-04

**Soundness:** 2
**Presentation:** 3
**Contribution:** 2
**Rating:** 5
**Confidence:** 4

**Summary:**

This paper introduces MS³M to predict vehicle trajectories for self-driving cars. The key innovation is using state space models (Mamba) instead of the usual transformer-based approaches that everyone's been using.

The system has three main parts:
 1. A temporal model that understands how things move over time
 2. A multi-stage setup that figures out how different objects interact in space
 3. A decoder that efficiently processes everything to make predictions

The authors tested it on the Argoverse 2 dataset and showed it holds up well against existing approaches.
Basically, they've shown you can get good results without relying on transformers, which might be useful for actual self-driving cars where speed and efficiency matter.

**Strengths:**

Novelty in using state space models for trajectory prediction
Shows you can match transformer performance with less compute
Method is simple to understand and implement
Solves a real problem (making self-driving predictions faster)

**Weaknesses:**

No code or detailed implementation provided - makes reproducibility impossible

Paper shows results only on Argoverse 2 benchmark
No cross-dataset validation
No real-world testing results
No robustness tests
Limited ablation studies

**Questions:**

Could you provide a concrete example showing how the FPN processes input sequences of different lengths (e.g., 10 vs 20 timestamps)?
What are the specific constraints on input sequence length, if any?
Please include a detailed diagram showing how temporal resolution changes at each FPN level, with specific dimensions and downsampling ratios.

What is the exact mathematical formulation used to predict anchor points between stages?
Are anchor points represented as discrete coordinates or continuous values?
Could you provide pseudocode for the anchor point computation algorithm?

Please detail the specific padding/masking operations used to handle varying numbers of agents
How are different numbers of lanes normalized or processed in the model architecture?
What is the maximum supported number of agents and lanes?
Could you provide an architecture diagram showing how variable-length inputs are processed through the network?

Could you provide cross-validation results on the nuScenes dataset, as it represents a different data distribution?
Please include robustness tests for varying weather conditions and occlusion scenarios

---

### Official Review · Reviewer_djNi · 2024-11-04

**Soundness:** 2
**Presentation:** 3
**Contribution:** 2
**Rating:** 3
**Confidence:** 1

**Summary:**

1) Introduce the Multi-Stage State Space Model (MS3M), to integrate a selective scanning mechanism into motion forecasting tasks.
2) The paper factors spatial and temporal tasks to two different models:  incorporates a Temporal Mamba Model to
capture fine-grained temporal information and a Spatial Mamba Model to model spatial interactions

**Strengths:**

1) The paper is well written and easy to understand.
2) A good overview of part of the existing work is summarized to help setup the motivations of the paper.
3) A good ablation analysis is presented to understand different parts of the contribution.

**Weaknesses:**

1) The paper is not comparing to a large portion of existing literature.
2) The factored reasoning over time and space can be achieved easily in the transformer domain using factored attention for example without need for the complicated architectural changed proposed here. There is not comparison to such transformer baselines, which makes it difficult to validate the claims made in the paper. [1]
3) Experimental section is weak with small number of results just on the Argoverse dataset.



[1] Wayformer: Motion Forecasting via Simple & Efficient Attention Networks

**Questions:**

1) Why no transformer based baselines have been included in the comparison?
2) Why WOMD is not used to strengthen the results section?

---

### Meta-Review · Area_Chair_cvV2 · 2024-12-08

**Metareview:**

The paper proposes multi-stage state space model for motion forecasting, given the motivation that the Transformer architectures have limitations in terms of efficiency. There are five reviews (typically four is standard practice. Note: AC noticed the additional review(s) might bring in more efforts during rebuttal and has considered this factor). The final review score were (3,3,3,5,5). The main concerns were:

- Lack of sufficient experiments on larger benchmarks other than Argoverse 2. This is the most raised concern by all reviewers.
- Lack of technical details.
- QCNet (current SOTA) works better than the proposed method. The contribution of "achieves superior performance while significantly reducing model size and latency" is over-claimed.
- Other factors (typos, incomplete comparision to other approaches, e.g. UniTraj, etc.)

There are consensus that the manuscript is not ready for publication. Investigation the new architecture is definitely a good direction to explore. AC agrees with all reviewers that sufficient experimetns need to be conducted to verify the proposed SSM architecture.

**Additional Comments On Reviewer Discussion:**

Out of five reviewers, two have given feedback after the rebuttal. Authors have been actively engaged in the rebuttal and provided additional experiments on the Argoverse 1 benchmark, with comparision to other SOTAs. Authors also commented on the discrepancy in performance with the QCNet.

Reviwers acknowledge most of the feedback and still believe the current version of the manuscript is not ready. In AC's perspective, there are still some important experiments to be accomplished before a ready version. For example, more sufficient verification on more datasets; missing technical details, etc.

Authors are strongly encouraged to polish the work and submit it at future venues.

---

### Decision · Program_Chairs · 2025-01-22

Reject